# Subtilisin-Involved Morphology Engineering for Improved Antibiotic Production in Actinomycetes

**DOI:** 10.3390/biom10060851

**Published:** 2020-06-03

**Authors:** Yuanting Wu, Qianjin Kang, Li-Li Zhang, Linquan Bai

**Affiliations:** 1State Key Laboratory of Microbial Metabolism, School of Life Sciences & Biotechnology, Shanghai Jiao Tong University, Shanghai 200204, China; yuantingwu1987@sjtu.edu.cn (Y.W.); qjkang@sjtu.edu.cn (Q.K.); 2Joint International Research Laboratory of Metabolic & Developmental Sciences, Shanghai Jiao Tong University, Shanghai 200240, China; 3College of Life Science, Tarim University, Alar 843300, China; zhang63lyly@sina.com

**Keywords:** actinomycetes, transcriptome, morphology engineering, mycelial fragmentation, subtilisin

## Abstract

In the submerged cultivation of filamentous microbes, including actinomycetes, complex morphology is one of the critical process features for the production of secondary metabolites. Ansamitocin P-3 (AP-3), an antitumor agent, is a secondary metabolite produced by *Actinosynnema pretiosum* ATCC 31280. An excessive mycelial fragmentation of *A. pretiosum* ATCC 31280 was observed during the early stage of fermentation. Through comparative transcriptomic analysis, a subtilisin-like serine peptidase encoded gene *APASM_4178* was identified to be responsible for the mycelial fragmentation. Mutant WYT-5 with the *APASM_4178* deletion showed increased biomass and improved AP-3 yield by 43.65%. We also found that the expression of *APASM_4178* is specifically regulated by an AdpA-like protein APASM_1021. Moreover, the mycelial fragmentation was alternatively alleviated by the overexpression of subtilisin inhibitor encoded genes, which also led to a 46.50 ± 0.79% yield increase of AP-3. Furthermore, *APASM_4178* was overexpressed in salinomycin-producing *Streptomyces albus* BK 3-25 and validamycin-producing *S. hygroscopicus* TL01, which resulted in not only dispersed mycelia in both strains, but also a 33.80% yield improvement of salinomycin to 24.07 g/L and a 14.94% yield improvement of validamycin to 21.46 g/L. In conclusion, our work elucidates the involvement of a novel subtilisin-like serine peptidase in morphological differentiation, and modulation of its expression could be an effective strategy for morphology engineering and antibiotic yield improvement in actinomycetes.

## 1. Introduction

The filamentous actinomycetes, with *Streptomyces* spp. as the major genus, are well known for their rich and diverse secondary metabolites, many of which have been developed into drugs (e.g., erythromycin, vancomycin) and agents for plant protection (e.g., avermectin, validamycin). On solid media, most of actinomycetes undergo a full cycle of morphological differentiation, initiated from spore germination, followed with the successive formation of vegetative mycelia, aerial mycelia and spore chains [1]. For industrial production of antibiotics in large scale, actinomycetes are usually subjected to submerged fermentation in liquid cultures. During submerged cultivation, the inoculated mycelia or mycelia germinated from spores start to form pellets, following which programmed cell death (PCD) occurs at the center, and new multinucleated mycelia develop inside or from the edge of the pellets to carry out antibiotic production [2]. In bioreactors, since the sizes and the densities of pellets are critical for nutrient and oxygen transfer, which are also triggering factors of PCD, mycelial morphology correlates with agitation, aeration, hydrodynamics and the yields of antibiotics [3]. Moreover, different antibiotic producing strains favor different morphologies. Whereas the pellet formation is required for nikkomycin production in *Streptomyces tendae* and erythromycin in *Saccharopolyspora erythraea* [4,5], it has negative effects on the production of tylosin in *Streptomyces fradiae* and nystatin in *Streptomyces noursei* [6,7]. Therefore, in order to optimize the production of antibiotics, morphology engineering of actinomycetes is needed. Classical strategies for morphology engineering, including adjustments of pH, temperature, medium composition, aeration, and agitation, have been implemented for the production of lipstatin, ε-poly-L-lysine, rapamycin, etc. [8,9,10]. However, these manipulations usually affect large pellets and have limited effect on small pellets [11].

Morphology engineering by genetic manipulation is more targeted and flexible. Recently, several protein families have been identified to play roles in the control of morphogenesis [12]. The first family are proteins involved in the formation of tip-organizing center (TIPOC) for apical growth and hypha branching, including the DivIVA, cytoskeletal protein Scy, cell-wall remodeling protein SsgA, the cellulose synthase-like CslA, and its cognate galactose oxidase-like GlxA [13,14,15,16,17]. The second family of proteins involved in the control of liquid-culture morphogenesis include a cyclic nucleotide-binding protein EshA and amine oxidase HyaS [18,19]. The third family of proteins, such as poly-β-1,6-*N*-acetylglucosamine (PNAG) synthases MatAB, are responsible for extracellular polymer synthesis [20]. Overexpression of SsgA was employed to obtain fragmented mycelia and fast growth of *Streptomyces venezuelae* and *Streptomyces coelicolor* M145 in submerged cultures, which had positive effects on the productions of zincphyrin IV and undecylprodigiosin, respectively [21,22]. When gene *hyaS* was deleted in *Streptomyces lividans*, the production of undecylprodigiosin was substantially improved along with outgrowing bunches of hyphae [19]. Moreover, the deletion of genes *matA* and *matB* in *S. lividans* led to dispersed mycelia and improved production of tyrosinase [20].

The antitumor agent ansamitocins, structurally similar to maytansines from *Maytenus serrata* [23], are produced by several genera of actinomycetes, including *Actinosynnema* [24,25], *Amycolatopsis* [26,27] and *Nocardiopsis* [28]. For large scale fermentation, ansamitocins are produced with *A. pretiosum* subsp. *auranticum* ATCC 31565, and ansamitocin P-3 (AP-3) is the major product and has the most potent antitumor activity [24]. Using maytansinol as a payload, the deacylated product of AP-3, antibody–drug conjugates have been recently developed as new strategy for cancer treatment [29], e.g., the Food and Drug Administration (FDA)-approved T-DM1 (Kadcyla^®^) for Human Epidermal Growth Factor Receptor 2 (HER2)-positive metastatic breast cancer [30]. Aided by the in-depth biosynthetic studies [31,32], the yield improvement of AP-3 has been intensively conducted through random mutagenesis and screening, process engineering, and pathway engineering, which includes the optimization of post-PKS modifications, enhancement of precursor supplies, improved gene expression, and so on [33,34,35,36,37,38,39]. However, the yield of AP-3 is still low and insufficient for supporting the clinical trials of drug leads in pipeline and subsequent clinical treatment.

Herein, an excessive mycelial fragmentation was found to be undesirable for the yield improvement of AP-3 in *A. pretiosum* subsp. *pretiosum* ATCC 31280. Through comparative transcriptomic analysis and gene inactivation, we identified a gene *APASM_4178*, coding for a subtilisin-like serine peptidase, to be responsible for the mycelial fragmentation. The transcription of *APASM_4178* was proven to be regulated by AdpA-like protein, and the mycelial fragmentation was alternatively alleviated by the overexpression of subtilisin inhibitor genes. Moreover, the overexpression of *APASM_4178* led to dispersed mycelia and substantially improved yields of salinomycin and validamycin in corresponding producing strains.

## 2. Materials and Methods

### 2.1. Strains, Plasmids, Media and DNA Techniques

The bacterial strains, plasmids and primers used in this study are listed in Appendix A. *A. pretiosum* ATCC 31280, the AP-3 producer, and its derivatives were cultured at 30 °C on YMG agar plates (0.4% yeast extract, 1.0% malt extract, 0.4% glucose, 2.0% agar (*w*/*v*), pH 7.2–7.3). For metabolites analysis, shake-flask fermentation was performed in 250-mL flask containing 30 mL of medium. The first seed medium (3.0% tryptone soya broth powder, 0.5% yeast extract, and 5.0% sucrose (*w*/*v*), pH 7.5) was inoculated with agar-grown mycelia and cultivated at 30 °C, 220 rpm for 24 h. Subsequently, the second seed medium (3.0% tryptone soya broth powder, 0.5% yeast extract, and 2.5% sucrose, 1.0% soluble starch (*w*/*v),* 0.05% isobutanol and 0.05% isopropanol (*v*/*v*), pH 7.5) was inoculated with 1 mL of the first seed culture and incubated for 24 h at 30 °C, 220 rpm. Fermentation medium (yeast extract 0.8%, malt extract 1.0%, sucrose 1.5%, soluble starch 2.5% (*w*/*v*), isobutanol 0.5%, and isopropanol 1.2% (*v*/*v*), pH 7.5) was inoculated with the second seed culture at 10% (*v*/*v*), and the fermentation continued for 8 days at 25 °C, 220 rpm. For the isolation of total DNA, *A. pretiosum* was cultivated in trypticase soy broth (TSB) [40] supplemented with 10.3% sucrose and 1% yeast extract at 30 °C for 24 h.

### 2.2. Growth Measurement and AP-3 Yield Determination

The mycelial growth was estimated by determining the mycelial dry weight. Mycelia from 1.0 mL of fermentation broth were collected by centrifugation at 12,000 rpm for 2 min, washed with 0.1 M HCl, and dried to constant weight at 65 °C for dry cell weight (DCW) measurement.

To quantify AP-3 yield, the supernatant of the fermentation broth was extracted with two volumes of ethyl acetate and evaporated. The residue was dissolved in methanol, passed through 0.22-μm filters, and subjected to high performance liquid chromatography (HPLC). HPLC analysis was operated on Agilent series 1260 (Agilent Technologies, Santa Clara, CA, USA) with an Agilent Eclipse Plus-C_18_ column (4.6 × 150 mm, 5 μm), using a previously described method [39].

### 2.3. Mycelial Morphology Observation

Mycelial morphology was observed using phase-contrast microscopy and laser confocal microscopy. After being washed twice with an equal volume of phosphate-buffered saline (PBS), 10 μL of mycelial suspension was pipetted onto a standard glass slide (25 × 75 mm) and covered with cover slide (24 × 24 mm). The images were captured with a digital camera CX41 mounted on the microscope (Olympus, Tokyo, Japan) with 400-fold magnification viewed under phase-contrast mode. For each sample, a minimum of five images were captured.

The dye FM 4-64FX (Molecular Probes, Inc., Eugene, OR, USA) with a concentration of 200 μM in Hank’s Balanced Salt Solution was used to dye the cell membrane. Equal volumes (10 μL) of FM 4-64FX solution and mycelia suspension were mixed and kept on ice for 1 min, and 10 μL of the mixture was pipetted on a clean slide and covered by a cover slip. Samples were observed with an SP8 STED laser scanning confocal microscope (Leica Microsystems, Buffalo Grove, IL, USA), equipped with an oil-immersed 100× objective and a charge coupled device (CCD) camera. Samples were excited at a wavelength of 558 nm and observed at an emission wavelength of 734 nm. Each mycelial sample was scanned at least for five times, and each scan was viewed in at least five fields.

### 2.4. Transcriptome Sequencing and Analysis of A. pretiosum ATCC 31280

The RNA for transcriptome sequencing was sampled at 15 h, 18 h and 24 h of fermentation. The total RNA was extracted using Trizol according to the manufacturer’s instruction. The transcriptome sequencing was conducted at Shanghai Biotechnology Corporation (Shanghai, China), and the expression level of each gene was calculated as Fragments Per Kilobase of exon per Megabase of library size (FKPM). The transcriptomic data have been deposited at GEO (Gene expression Omnibus) with an accession number of GSE151010. With the collected data, the transcription levels between 15, 18, and 24 h were compared. Genes with the lowest transcription at 18 h were selected.

### 2.5. Transcription Analysis by Quantitative Real-Time PCR

Mycelia in fermentation media were collected at different times. Total RNA was extracted with Redzol reagent according to the procedures described by the manufacturer (SBS Genetech, Beijing, China). The purity/concentration was assessed using Nanodrop 2000 (ThermoFisher, Waltham, MA, USA). Total DNA was removed by DNase I (ThermoFisher), and reverse transcription was carried out using RevertAid^TM^ H Minus First Strand cDNA Synthesis Kit (ThermoFisher). The transcription of target genes was determined by quantitative real-time PCR (qRT-PCR) on a 7500 Fast Real-time PCR System (Applied Biosystems, Carlsbad, CA, USA) with One-Step PrimerScript^TM^ RT-PCR Kit according to the manufacturer’s procedure (Takara, Dalian, China). The transcription of target genes was internally normalized to the housekeeping gene *hrdB* and quantified by the 2^−ΔΔCT^ method [41]. Each reaction was performed in triplicate.

### 2.6. Inactivation and Trans-Complementation of Gene APASM_4178

To inactivate *APASM_4178*, the genomic DNA of *A. pretiosum* ATCC 31,280 (GenBank accession no. CP029607.1) was used as PCR template. A 1.38-kb XbaI-EcoRI fragment of the left flanking region and a 1.45-kb EcoRI-HindIII fragment of the right flanking region were ligated with XbaI/HindIII-digested plasmid pJTU1278 to generate plasmid pLQ855. The recombinant plasmid was transferred into *Escherichia coli* ET12567(pUZ8002) and then introduced into *A. pretiosum* ATCC 31,280 by intergeneric conjugation, and thiostrepton-resistant (Thio^R^) exconjugants were selected. Subsequently, *APASM_4178*-deletion mutant WYT-5 was selected from the initial Thio^R^ exconjugants after two rounds of nonselective growth. Using primers 4178-ver-F/R, a 0.5-kb fragment was amplified by PCR from the mutant WYT-5, whereas a 3.7-kb fragment was amplified from *A. pretiosum* ATCC 31280. Other genes were similarly inactivated.

For the trans-complementation of mutant WYT-5, gene *APASM_4178* was amplified with primers 4178-F/R and cloned in NdeI/EcoRI-digested pLQ856 under the control of *kasOp** promoter. The recombinant plasmid pLQ874 was transferred into *E. coli* ET12567(pUZ8002) and then introduced into mutant WYT-5 by intergeneric conjugation. Apramycin-resistant (Apr^R^) exconjugants were selected and named as WYT-13.

### 2.7. Overexpression of Subtilisin Inhibitor Genes

For the overexpression of gene *APASM_3064* in *A. pretiosum* ATCC 31280, *APASM_3064* was amplified with primers 3064-F/R and cloned in NdeI/EcoRI-digested pLQ856 under the control of *kasOp** promoter. The recombinant plasmid pLQ869 and control plasmid pLQ856 were individually transferred into *E. coli* ET12567(pUZ8002) and then integrated into the *attB* site of *A. pretiosum* ATCC 31,280 by intergeneric conjugation. Apramycin resistant (Apr^R^) exconjugants were selected. Derivative strain overexpressing gene *APASM_6209* was similarly constructed.

### 2.8. Heterologous Overexpression of the AdpA-Like Protein APASM_1021

Gene *APASM_1021* was amplified with primers 1021-F/R and cloned in HindIII/NdeI-digested pET28a. The recombinant plasmid pLQ864 was transformed into *E. coli* BL21(DE3) for protein overexpression. Colonies were picked and cultivated at 37 °C in LB medium containing 50 μg/mL kanamycin until an optical density at 600 nm (OD_600_) of 0.6–0.8 was reached. The culture was then induced with 0.1 mM isopropyl-β-D-thiogalactoside (IPTG) and incubated further for 24 h at 16 °C. The induced cells were harvested, resuspended in 50 mM Tris-HCl (pH 7.5), and disrupted by ultrasonication. For the purification of His-tagged APASM_1021, cells were harvested by centrifugation, resuspended with buffer A (20 mM Tris-HCl, 300 mM NaCl, pH 8.0) and sonicated for 30 min on ice, followed by centrifugation for 40 min at 12,000 rpm and 4 °C. Proteins from the supernatant were purified with Ni Sepharose^TM^ 6 Fast Flow (GE Healthcare Life Sciences, Marlborough, MA, USA). The eluted His-tagged APASM_1021 was determined by 12% sodium dodecyl sulfate-polyacrylamide gel electrophoresis (SDS-PAGE).

### 2.9. Electrophoretic Mobility Shift Assays

A carboxyfluorescein (FAM)-labeled DNA probe P4178 was amplified using primers 4178p-F-FAM/4178p-R (Appendix A) and genomic DNA from *A. pretiosum* ATCC 31,280 as template. In a total volume of 20 μL, 100 ng FAM-labeled probes were incubated for 30 min at 30 °C with different amounts of purified His-tagged APASM_1021 in 1× binding buffer containing 10 mM Tris-HCl (pH 7.5), 1 mM EDTA, 100 mM KCl, 5% (*v*/*v*) glycerol, 0.1 DTT, and 0.01 mg/mL BSA [42]. The resulting DNA-protein complexes were then subjected to electrophoresis on 10% native polyacrylamide gels for 45 min at 4 °C [43]. The band shift was analyzed by XcitaBlue^TM^ Conversion Screen Kit (Bio-Rad, Hercules, CA, USA).

## 3. Results

### 3.1. An Early and Severe Mycelial Fragmentation in Liquid Cultures of A. pretiosum ATCC 31280

In shake flask fermentation inoculated from a seed culture, the ansamitocin-producing *A. pretiosum* ATCC 31280 gained exponential growth in the first 24 h, reached stationary phase at 48 h, and maintained a relatively constant biomass (14.95 ± 0.64 mg/mL) until day 6 (Figure 1A), albeit much lower than that of most actinomycetes. However, when we measured the biomass at three-hour intervals in the first two days, a sudden decrease was observed at 18 h, followed by an increase from 21 to 24 h. Interestingly, this decrease-then-increase pattern occurred again from 24 to 33 h (Figure 1B).

For a better understanding of this phenomenon, mycelia of the fermentation broth were collected at 15, 18, and 24 h and checked under both phase-contrast light microscope (PCM) and confocal laser scanning microscope (CLSM). Stained with membrane-binding dye FM 4-64FX for CLMS, a severe fragmentation with short hyphae and less biomass was observed with the 18-h sample, whereas the mycelia of the 15 and 24-h samples formed thick clumps (Figure 1C). Moreover, the colony-forming units (CFUs) of each sample were quantified, and 2.0 × 10^7^, 1.0 × 10^7^, and 32.5 × 10^7^ CFUs/mL were found to be present in the 15, 18, and 24-h samples, respectively (Figure 1D), suggesting a concurrent cell lysis along with the mycelial fragmentation at 18 h, which most likely led to the biomass decrease. In addition, five different media were used (Appendix A), and the mycelial fragmentation occurred in all of them around 18 h (data not shown), indicating that this unique fragmentation feature is genetically determined.

### 3.2. Comparative Transcriptomic Analysis Identified a Subtilisin-Like Protease Gene Responsible for the Mycelial Fragmentation

In order to identify genes involved in the early mycelial fragmentation, the total RNAs of mycelia collected at 15, 18, and 24 h were extracted and subjected to transcriptome sequencing using RNA-seq technology. We assumed that genes involved in the fragmentation have the lowest transcription at 18 h and therefore higher transcription at 15 h and 24 h. According to this criterion, comparative analysis was subsequently performed among these three transcriptomes, and 12 genes were identified from a total of 7038 genes of the *A. pretiosum* ATCC 31,280 genome, with a ≥ 2-fold higher transcription at 15 h and 24 h than that at 18 h (Appendix A). The transcription of these 12 genes was further verified through quantitative real-time PCR (qRT-PCR), among which 10 were found to keep the similar patterns with those obtained by transcriptome analysis. Genes *APASM_1687* and *APASM_4084* had continuously increased transcription from 15 to 24 h (Appendix A).

These 12 candidate genes were individually deleted through double crossover homologous recombination to investigate any involvement in the fragmentation (Appendix A). Whereas the deletions of 11 genes had no noticeable effect on the fragmentation (Appendix A), the deletion of gene *APASM_4178*, coding for a subtilisin-family serine peptidase, led to a delayed fragmentation by 30 h, from the original 18 h in the wild-type to 48 h in the mutant WYT-5 (Figure 2A). Moreover, the *APASM_4178* deletion mutant WYT-5 had a 26.99% improved biomass of 22.77 mg/mL at day 4 and a 43.65% increased AP-3 yield (56.64 mg/L) compared with those of the wild-type (Figure 2B,C). Further confirmation came from the trans-complementation of the mutant WYT-5 with a cloned *APASM_4178* under the control of *kasOp** promoter on a ΦC31-derived integrative plasmid, which resulted in a recovered mycelial fragmentation at 18 h in mutant WYT-13, similar to that of the wild-type *A. pretiosum* ATCC 31280 (Figure 2D).

### 3.3. The Transcription of APASM_4178 and Mycelial Fragmentation is AdpA-Like-Dependent

The recurrent fragmentation of *A. pretiosum* ATCC 31280 in early growth phase suggests an involvement of quorum sensing-like regulation (Figure 1B). The γ-butyrolactone system is the primary quorum sensing system in actinomycetes, and the widespread A-factor-dependent proteins (AdpA) mediate the regulatory signals of various γ-butyrolactone molecules and exert pleiotropic regulation on morphology differentiation and secondary metabolism [44,45]. Therefore, the upstream region of *APASM_4178* was analyzed, and two conserved AdpA-binding motifs, 5′-TGACGGGGAG-3′ and 5′-CGCGCCGCCA-3′ (in reversed orientation), were identified (Figure 3A) [46].

Moreover, four genes, namely *APASM_1021*, *APASM_3332*, *APASM_4306*, and *APASM_5462*, were found to be annotated as AdpA-like-coding genes in the genome of *A. pretiosum* ATCC 31280. However, they share a relatively low sequence identity between 39.62 and 51.46% (Appendix A). Subsequently all four genes were individually deleted through homologous recombination in *A. pretiosum* ATCC 31280, and the transcription of *APASM_4178* in each mutant was analyzed through qRT-PCR. In the *APASM_1021* deletion mutant WYT-15, the transcription of *APASM_4178* was substantially reduced, whereas its transcription in the mutants of *APASM_3332*, *APASM_4306*, and *APASM_5462* remained as high as that of the wild-type during the first 48-h cultivation (Figure 3B) (Appendix A). Accordingly, the mycelial morphology was examined under phase-contrast light microscope. As expected, the mycelial fragmentation occurred at 18 h in the wild-type but not in mutant WYT-15 (Figure 3C), suggesting that *APASM_1021* is involved in the expression of *APASM_4178*. However, the production of AP-3 is abolished (data not shown) in WYT-15, which could be explained by the pleiotropic regulatory roles of AdpA-like proteins in actinomycetes [47].

Furthermore, an electrophoretic mobility shift assay (EMSA) was performed with purified His-tagged APASM_1021 and a 131-bp 5′-FAM-labelled DNA probe P4178, containing the two conserved AdpA-binding motifs within the putative promoter region of *APASM_4178* (Figure 3A). APASM_1021 was found to bind the labeled probe P4178 at concentrations above 0.5 μM. When 10-fold of unlabeled probe P4178 was added to the binding mixture, the binding was competitively reversed, whereas a supplementation of 10-fold of nonspecific poly(dI-dC) had no effect (Figure 3D). The above-described results confirmed a specific regulation of the AdpA-like protein APASM_1021 on the expression of the subtilisin-like peptidase APASM_4178.

### 3.4. Overexpression of Subtilisin Inhibitors Alleviated Mycelial Fragmentation and Increased AP-3 Production

Subtilisin-like serine peptidases are widely spreaded in actinomycetes, and it is believed to be involved in morphology differentiation and other protein maturation processes. The activities of subtilisins are usually modulated by cognate subtilisin inhibitors (SSI) through specific binding [48,49]. Therefore, the genome sequence of *A. pretiosum* ATCC 31280 was analyzed and two SSI genes, *APASM_3064* and *APASM_6209*, were identified. Since the presence of subtilisin-like APASM_4178 causes the mycelial fragmentation, these two SSI genes were individually overexpressed in *A. pretiosum* ATCC 31280 to test if there is an inhibition on the activity of APASM_4178 and consequent alleviation of mycelial fragmentation. As expected, the overexpression of each gene under the control of a strong *kasOp** promoter led to a disappearance of mycelial fragmentation during the first 48-h cultivation (Figure 4A,B). Moreover, the overexpression of *APASM_3064* led to 34.54% and 47.28% increase of biomass at day 5 and AP-3 yield, respectively, while the overexpression of *APASM_6209* led to 42.60% and 45.71% increase of biomass at day 5 and AP-3 yield, respectively, compared with those of the wild-type integrated with vector pBLQ856 (Figure 4C,D). However, the integration of ΦC31-derived plasmids caused decreased AP-3 yield for unknown reason.

### 3.5. Overexpression of APASM_4178 Led to Dispersed Mycelia and Improved Antibiotic Yields in Streptomyces Strains

Whereas *A. pretiosum* ATCC 31280 undergoes mycelial fragmentation at early stage of cultivation and has less biomass accumulation, most *Streptomyces* strains have robust growth and form large dense pellets consisting of interconnected hyphae, which usually limit the transfer of nutrients and oxygen and consequent antibiotic production [12]. In an attempt to reduce to the size of pellets, the subtilisin-like gene *APASM_4178* under the control of promoter *kasOp** was introduced into salinomycin-producing *Streptomyces albus* BK 3-25 [50] and validamycin-producing *Streptomyces hygroscopicus* TL01 [51]. Interestingly, the overexpression of *APASM_4178* in both strains indeed led to dispersed mycelia (Figure 5A,B). Consequently, the yield of salinomycin was increased by 33.80%, from 17.99 to 24.07 g/L, and that of validamycin was increased by 14.94%, from 18.67 to 21.46 g/L (Figure 5C,D).

## 4. Discussion

The mycelial fragmentation of *A. pretiosum* in liquid cultures has long been noticed [52] and even utilized for efficient selection of mutants through double crossover recombination [31]. However, when yield improvement was considered for future large-scale fermentation, the accompanying low biomass (14.95 mg/mL) with the mycelial fragmentation became a serious limit (Figure 1A). In order to eventually alleviate the fragmentation, RNA-seq transcriptome analysis was performed with samples collected with 3 or 6-h intervals, and the involved gene *APASM_4178* was identified among 12 genes with the lowest transcription at 18 h of cultivation, the very moment of excessive fragmentation (Appendix A). Indeed, the deletion of *APASM_4178* led to an alleviated fragmentation, 26.99% improved biomass and 43.65% increased AP-3 yield (Figure 2A–C). Our work displays the accuracy and efficiency of comparative transcriptome analysis in the identification of targets for genetic engineering, as also shown in the titer improvements of actinorhodin in *S. coelicolor* and of spiramycin in *Streptomyces ambofaciens* [53,54].

Gene *APASM_4178* is 3,276 bp and predicted to code for a 1091-aa 120.0-kDa multi-domain protein, composed of a 150-aa N-terminal pro-peptide region, a 236-aa subtilisin-like serine peptidase domain, and a 616-aa C-terminal region. A BLAST search using APASM_4178 as a query identified more than 100 homologs with sequence identities higher than 40% and sequence coverage higher than 91%. Moreover, most of them are from rare actinomycetes, including *Actinosynnema*, *Saccharothrix*, *Lentzea*, *Saccharomonospora*, *Actinophytocola*, and *Micromonospora*. Subtilisins are extracellular alkaline serine peptidases with a catalytic triad, consisting of Asp, His, and Ser, and are found to be widely involved in morphology differentiation and other protein maturation processes [48,49]. Different from APASM_4178 and its homologs, typical subtilisins from actinomycetes are smaller than 65.0 kDa and contain an N-terminal pro-peptide and the serine peptidase domain [55]. Due to the presence of an extra 616-aa C-terminal region in APASM_4178, its proteolytic activity was measured and compared with APASM_6525, a 519-aa subtilisin-like serine peptidase from *A. pretiosum* ATCC 31280, and SCO1355, a 537-aa subtilisin-like serine peptidase from *S. coelicolor* A3(2) [49]. The results showed that APASM_4178 has similar peptidase activity to those of APASM_6525 and SCO1355, indicating that it is indeed a subtilisin-like serine peptidase (Appendix A). However, the function of the large C-terminal region is still unknown and needs further investigation. Interestingly, the large C-terminal region of a cell envelope-located serine proteinase from *Lactococcus lactis* SK11 was proposed to be involved in anchoring of the enzyme in the cell membrane [56].

In actinomycetes, the expression of subtilisin and its cognate SSI is usually under the control of AdpA, the pleiotropic regulatory protein in the A-factor signaling cascade [57]. Even though APASM_4178 is substantially larger than typical subtilisins, its expression is similarly regulated by AdpA-like protein, as proved by in vivo deletion of AdpA-like coding gene and in vitro EMSA (Figure 3). The early and periodic occurrence of mycelial fragmentation in *A. pretiosum* ATCC 31280 actually resembles quorum sensing behavior responding to A-factor-like signal molecules. However, searching for A-factor and butanolide biosynthetic genes in the genome of *A. pretiosum* ATCC 31280, using *scbA* from *S. coelicolor* and *sabA* from *Streptomyces ansochromogenes* as queries, failed to retrieve any homologous genes, suggesting the presence of structurally different γ-butyrolactone in *A. pretiosum* ATCC 31280. Moreover, the alleviated mycelial fragmentation still happens after 48 h in the *APASM_4178* deletion mutant WYT-5. When SSI genes were overexpressed in *A. pretiosum* ATCC 31280, the mycelial fragmentation did not occur, indicating the involvement of other subtilisin-like serine peptidase(s) in this process, which needs further investigation in the future.

In previous works for morphology engineering of actinomycetes, the cellulose synthase-like gene *cslA* [16], galactose oxidase-like gene *glxA* [17], PNAG synthase genes *matA* and *matB* [20], and amine oxidase gene *hyaS* [19] were deleted on one hand, and the cell-wall remodeling protein gene *ssgA* was overexpressed on the other hand. In our work, the large subtilisin-like serine peptidase gene *APASM_4178* was used for the first time to engineer *Streptomyces* spp. for dispersed mycelia and improved antibiotic yields, which could be applied with many of other antibiotic-producing actinomycetes, which form dense pellets in submerged culture. In addition, the overexpression of SSI in *A. pretiosum* ATCC 31280 resulted in an improved AP-3 yield higher than that of the *APASM_4178* deletion mutant. This strategy is applicable to some rare actinomycetes with mycelial fragmentation, especially those with homologous genes of *APASM_4178*.

## 5. Conclusions

This study identified a novel subtilisin-like serine peptidase gene by comparative transcriptome analysis in *A. pretiosum* ATCC 31280, and the deletion of this gene substantially reduced the mycelial fragmentation, increased biomass accumulation, and improved AP-3 yield. As a subtilisin-encoding gene, its transcription was controlled by a specific AdpA-like protein. Through the overexpression of subtilisin-binding SSI, the mycelial fragmentation was completely eliminated. The improved yields of AP-3, validamycin, and salinomycin along with altered morphologies of the producing strains highlighted new strategies for morphology engineering in actinomycetes.

## Figures and Tables

**Figure 1 biomolecules-10-00851-f001:**
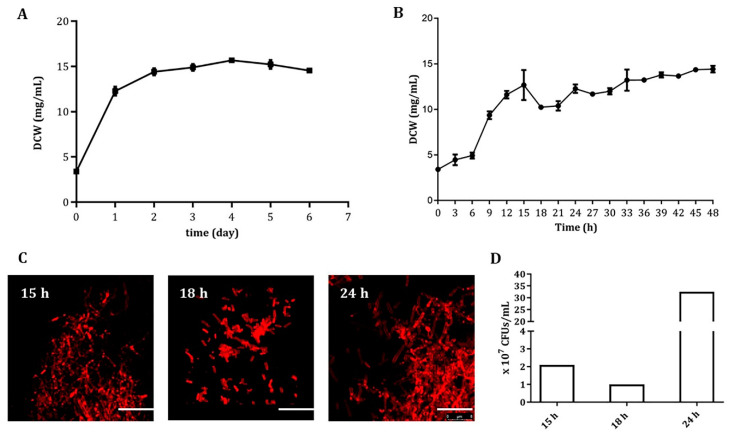
The growth and mycelial fragmentation of *A. pretiosum* ATCC 31280 in submerged cultures. (**A**) Growth curve of *A. pretiosum* ATCC 31280 during 6-day fermentation. Samples were taken daily. Error bars: mean ± standard deviation (SD; *n* = 3 biological replicates); (**B**) Growth of *A. pretiosum* ATCC 31280 during the first 48-h fermentation. Samples were taken every 3 h. Error bars: mean ± SD (*n* = 3 biological replicates); (**C**) Confocal micrographs of *A. pretiosum* ATCC 31280 at 15, 18 or 24 h. Scale bar: 10 μm. Magnification, 2500×; (**D**) Colony forming units (CFUs) of cultures at 15, 18 or 24 h.

**Figure 2 biomolecules-10-00851-f002:**
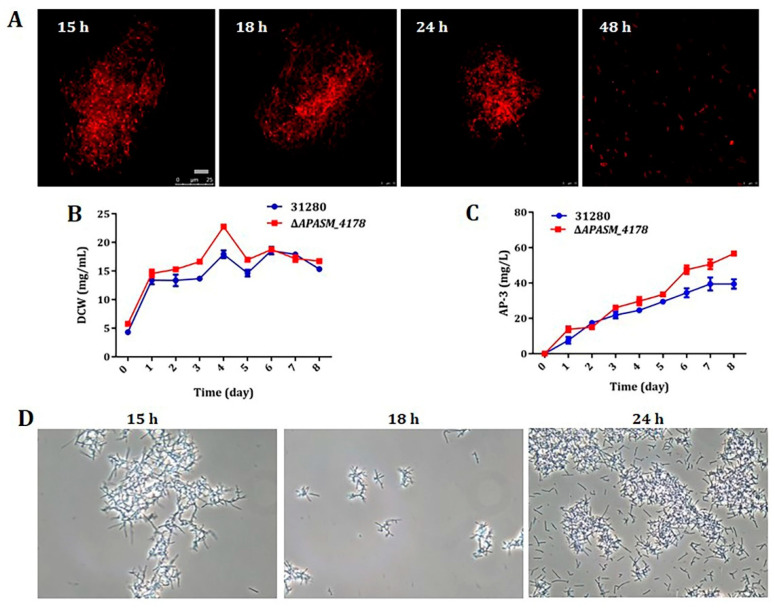
Deletion of subtilisin-like gene *APASM_4178* led to delayed mycelial fragmentation, increased growth and AP-3 yield. (**A**) Confocal micrographs of the *APASM_4178*-deleted mutant WYT-5. Scale bar: 10 μm. Magnification, 2500×; (**B**) Growth comparison between the wild-type and the mutant WYT-5. Error bars: mean ± SD (*n* = 3 biological replicates); (**C**) AP-3 yield comparison between the wild-type and the mutant WYT-5. Error bars: mean ± SD (*n* = 3 biological replicates); (**D**) Phase-contrast micrographs of WYT-5 complemented with cloned *APASM_4178*. Scale: 5 μm. Magnification, 400×.

**Figure 3 biomolecules-10-00851-f003:**
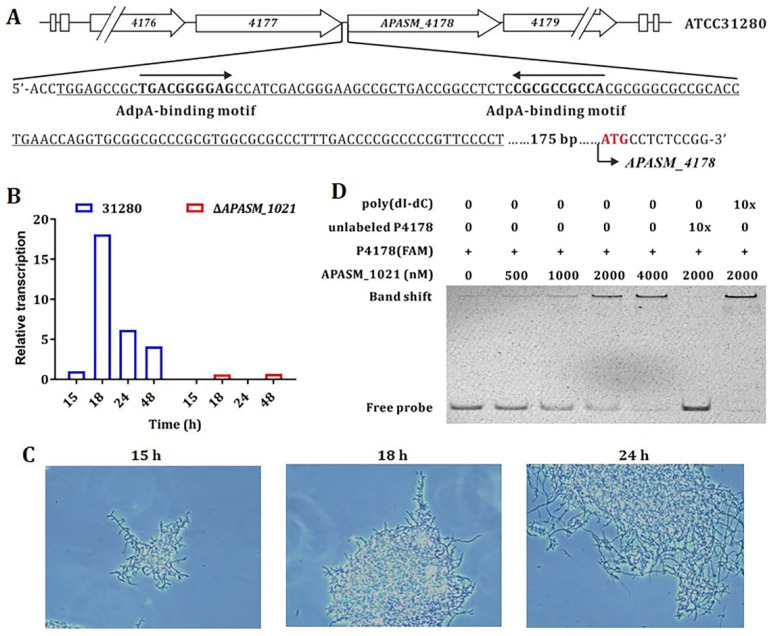
The transcription of *APASM_4178* is regulated by an AdpA-like protein APASM_1021. (**A**) Conserved AdpA-binding motifs in the upstream region of *APASM_4178*. Underlined sequences refer to the 131-bp region covered by P4178, the 5′-FAM-labeled probe for electrophoretic mobility shift assay (EMSA). The start codon of *APASM_4178* is shown in red; (**B**) The transcription of *APASM_4178* in the wild-type and the *APASM_1021*-deleted mutant WYT-15. The transcription level of *APASM_4178* in the wild-type at 15 h is set as 1; (**C**) Phase-contrast micrographs of the *APASM_1021*-deleted mutant WYT-15. Scale: 5 μm. Magnification, 400×. (**D**) EMSA of the specific binding between the upstream region of *APASM_4178* and purified APASM_1021. P4178 is a 131-bp 5′-FAM-labeled probe.

**Figure 4 biomolecules-10-00851-f004:**
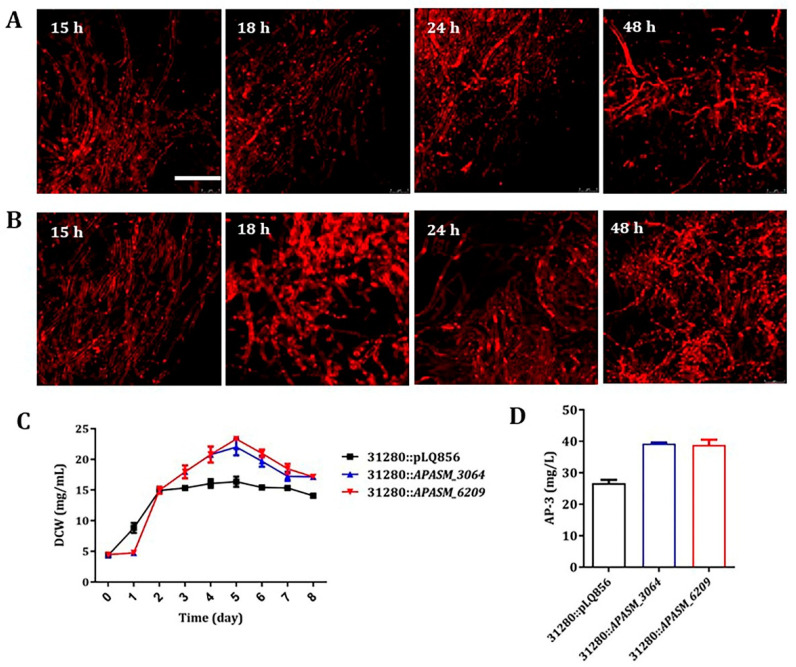
Overexpression of subtilisin inhibitor (SSI) genes in *A. pretiosum* ATCC 31280 alleviated mycelial fragmentation. (**A**) Confocal micrographs of mutant with SSI gene *APASM_3064* overexpressed. Scale bar: 10 μm. Magnification, 2500×; (**B**) Confocal micrographs of mutant with SSI gene *APASM_6209* overexpressed. Scale bar: 10 μm. Magnification, 2500×; (**C**) Growth comparison between 31280::pLQ856, the wild-type integrated with vector plasmid pLQ856, and mutants with *APASM_3064* or *APASM_3064* overexpressed. Error bars: mean ± SD (*n* = 3 biological replicates); (**D**) Yield comparison between the control strain 31280::pLQ856 and mutants with *APASM_3064* or *APASM_6209* overexpressed. Error bars: mean ± SD (*n* = 3 biological replicates).

**Figure 5 biomolecules-10-00851-f005:**
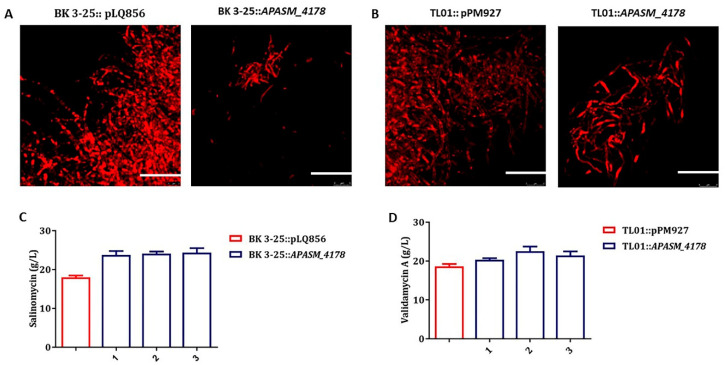
Overexpression of the subtilisin-like serine peptidase gene *APASM_4178* in salinomycin-producing *S. albus* BK 3-25 and validamycin-producing *S. hygroscopicus* TL01. (**A**) Effect of *APASM_4178* overexpression on the mycelial morphology of *S. albus* BK 3-25. Scale bars: 10 μm. Magnification, 2500×; (**B**) Effect of *APASM_4178* overexpression on the mycelial morphology of *S. hygroscopicus* TL01. Scale bars: 10 μm. Magnification, 2500×; (**C**) Effect of *APASM_4178* overexpression on salinomycin yield of *S. albus* BK 3-25. pLQ856: vector plasmid. Three mutant strains were randomly selected. Error bars: mean ± SD (*n* = 3 biological replicates); (**D**) Effect of *APASM_4178* overexpression on validamycin yield of *S. hygroscopicus* TL01. pPM927, vector plasmid. Three mutant strains were randomly selected. Error bars: mean ± SD (*n* = 3 biological replicates).

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
