# Peer review of "Subtilisin-Involved Morphology Engineering for Improved Antibiotic Production in Actinomycetes"

_biomolecules, 2020, doi:10.3390/biom10060851_

Round 1
Reviewer 1 Report
This is well planned and executed study that reports novel genetic circuit that controls morphology of rare (yet practically important) actinomycete in submerged culture. Authors went on to show that manipulations of this circuit impact titers of natural product in native strain, as well as in two industrial streptomycetes. I think therefore that findings of this work will have impact on how researchers approach the same problem in streptomycetes and other actinobacterial genera. A few minor comments are as follows. First, the way authors describe APASM_1021 operator sites is a bit confusing. The first one (CGCGCCGCCA) is actually recognized on opposite DNA strand, as judged from the figure 3, and I would note that in the text. Second, I disagree that APASM_1021 can be qualified as AdpA. Yes, the operator is similar, but it does not seem to show any orthology to Streptomyces AdpAs in reciprocal BLASTP searches. Does APASM_1021-encoding genome region have synteny to typical AdpA-encoding segments of streptomyces? I suggest to call APASM_1021 an AdpA-like protein, to avoid the misunderstanding (as a side note, this work is therefore one of pioneering studies of ApdA paralogs - we know nothing about them, and this work points possible research venues). Third, did the authors check the presence of butenolide biosynthetic geens in A. pretiosum? Butenolides were recently shown to control AdpA in nikkomycin producer; this might be added to Discussion section.
Reviewer 2 Report
This work by Wu et al represents an interesting topic of using genetic manipulation approach to engineer / control the morphology of actinomycetes for increasing the production of secondary metabolites. They systematically characterized the subtilisin-like serine peptidase encoded gene from
Actinosynnema pretiosum ATCC 31280. Morphology engineering could be one useful way to increase the yield of secondary metabolites in actinomycetes.
Some concerns should be addressed before publication:
1. Please find a native speaker to proofread your manuscript, the writing should be improved.
2. Line 13, …for antibiotic production… I would like to suggest changing “antibiotic” to “secondary metabolites”.
3. Line 14, change “…is produced by Actinosynnema pretiosum ATCC 31280 as a secondary metabolite…” to “is a secondary metabolite produced by Actinosynnema pretiosum ATCC 31280”
4. Line 15, change “An excessive mycelial fragmentation was…” to “An excessive mycelial fragmentation of Actinosynnema pretiosum ATCC 31280 was…”
5. Line 16, change “a subtilisin-like serine peptidase gene” to “a subtilisin-like serine peptidase encoded gene”
6. Line 18, deleted should be deletion.
7. Line 20, change “subtilisin inhibitor” to “subtilisin inhibitor encoded”
8. Line 22, overexpression should be overexpressed.
9. Lines 34-35, delete “for clinical treatment”
10. Line 37, put a comma behind spore germination
11. In Figure 1, please provide the magnification
12. Line 199, I highly suggest to at least list the names of five different media, which can give readers more information on how different these media are.
13. Line 209, results subsection of Comparative Transcriptomic Analysis, used RNA-Seq, but it was not included in Methods section. Please fulfill this information.
14. Lines 213-214, I did not fully understand this sentence: “Due to a temporal lag between gene transcription and the corresponding phenotype, 213 genes involved in the fragmentation are supposed to have the lowest transcription at 18 h and higher 214 transcription at 15 h and 24 h”. Please make it clear.
15. Line 218, Where is the full result of RNA-Seq? And related analysis? How were results in table S3 obtained?
16. Line 222, These 12 candidate genes were individually deleted. I did not count all constructed mutants in table S1. Please provide such strain information in table S1 and their related primers in table S2.
17. For comparison prospective, I suggest including the result of the 12th mutant in figure S3 as well.
18. For consistency prospective, I suggest using Confocal micrographs to replace the optical microscope photos in figure 2d
19. Line 252, remove “first”
20. Line 253, change “mutant WYT-15 with APASM_1021 deleted” to “the APASM_1021 deleted mutant WYT-15 ”
21. Lines 259-260, However, the production of AP-3 is abolished (data not shown) in WYT-15, which could be explained by the pleiotropic regulatory roles of AdpA in actinomycetes. It might be the off-target effects of WYT-15 construction, the genetic manipulation process broke the AP-3 biosynthetic pathway. How can you rule this possibility out?
22. Line 267, change “experiments” to “results”
23. Line 281, change “are wide spread” to “are widely spreaded
24. Line 281, change “and believed” to “and it is believed”
25. Lines 281-283, please rephrase the long sentence, try to use short and clear sentences.
26. In figure 4c and 4d, please also include the wt 31280 without bearing any plasmids.
27. In figure 45c and 5d, please also include the wt without bearing any plasmids.
Round 2
Reviewer 2 Report
Most of my concerns have been properly addressed. The reasons for other unchanged ones also convinced me, therefore, I suggest accepting as it is.